# Benchmarks Old and New: How to compare domain independence for cost-optimal classical planning?

**Ionut Moraru and Stefan Edelkamp**
Informatics Department, Kings College London
Strand Campus, Bush House, 30 Aldwych, London, WC2B 4BG
ionut.moraru@kcl.ac.uk and stefan.edelkamp@kcl.ac.uk

## Abstract

Domain independence is one of the main features of automated planning. Planners, in the context of cost-optimal classical planning, are developed with the intent of solving any type of problem that can be formulated in PDDL. We then compare planners by the number of problem instances they solve on a set of benchmarks, one point for each problem solved. However, does solving the most problems automatically result in having the best domain-independent planner?

In this paper, we compare the best performing, non-portfolio planners from the cost-optimal classical track of the International Planning Competition (IPC) 2018 on the complete set of benchmarks from the previous two competitions (2011 and 2014) and on a subset of the competitions from before 2011. Results show that, as the number of problems for each domain varies, current way of comparing planners (total coverage) can be biased towards the planners that perform the best in the domains with the most instances, but once we normalize those results, we can get a better picture for which technique is the most domain independent.

## 1   Introduction

Marvin Minsky classified Artificial Intelligence into five areas (1961), one of them being *planning*. Planning is the discipline that has the task of coming up with a sequence of actions that, starting from an initial state, will achieve a goal.

A technique that has appeared in the 80's and which has been giving really good results is the creation of solver software for resolving well-defined mathematical models, i.e. Constraint Satisfaction, Linear Programming, etc. (Geffner 2014). They are a general type of software, created for computing the solution of any problem specified in its input modeling language. In the field of AI Planning, this type of solvers are called *planners*, most of them using as an input PDDL (McDermott 1998; Fox and Long 2003).

Domain-independence is one of the qualities that planning as an area of research strives to achieve. What this means is that, as long as the problem is specified properly in PDDL, a planner should be able to give a valid plan as an output (Howey, Long, and Fox 2004). This is an extremely ambitious goal, especially when taking into consideration the complexity of a planning task (Bäckström and Nebel 1995;

Bylander 1994). Nonetheless, current planners are able to solve a large variety of problems from very different domains (from solving the Rubik's cube and Solitaire games, to Logistics and path-finding problems just as an example) which cannot be described as anything but incredible.

The current way of comparing planners in the setting of domain-independent classical planning is by seeing how many problems each planner can solve. Each problem validly solved gives towards the planner one point. The planner with the most *points* can be considered the *best* on the tested benchmarks. While this way has its values, having a planner that solves the most problems is a feat that should be celebrated, in our view this does not capture the complete picture of domain-independence.

In this paper, we will be arguing that classical planners should be compared in more ways than just how many problems they can solve. We will be taking the four best performing, non-portfolio, cost-optimal planners from the 2018 International Planning Competition and compare them on a larger set of problems. We have seen that, because of the different number of problems for each domain, some domains are more *important* than others when comparing just the total coverage over the benchmark set. We argue for the introduction of a normalized domain coverage metric, which would alleviate this issue and would be more representative for comparing planners in domain-independent planning.

## 2   International Planning Competition

The International Planning Competition (IPC) has been a great driver for progress of research and has brought forth many novel techniques and planning technologies since it's inception in 1998. Organized together with the International Conference of Automated Planning and Scheduling, it has build an identity synonymous with state-of-the-art for planning in any of its forms (in 2018 we had Classical, Probabilistic and Temporal tracks).

At the beginning, events were held every two years, as the planning research in the modern sense was in developing fast, but recently, as the benchmark sets available were larger and more planners were broadly available for inspection, advances have slowed down. Competitions are now organized every 3-4 years, giving time for researchers to advance the field and implement any new idea.

## Importance

IPC have brought a lot of benefits for the subject as a whole, first and foremost with PDDL, the high-level modeling language that has now become an informal standard input for most planners. PDDL was used from the first edition, bringing all the new versions and new features for one of the subsequent editions. As all the domains and problems are formulated using this modeling language, almost all modern planners are built now to support one of the versions of PDDL and more recently RDDL for Probabilistic Planning (Sanner 2010).

Continuing on the topic of benchmarks, each edition published either completely new or reinterpretations of domains with new problems, increasing number and diversity of available benchmarks for the planning community. This gives planner developers a more complete way of evaluating their systems.

Finally, competitions in any field bring together any community and it manages to evolve a field. Comparing in a closed environment a vast number of planners, each approaching problems in a different way, has the benefit of putting head-to-head each method without bias. As the benchmarks are not know prior to the planner submission, developers of said systems need to focus on creating domain-independent planners, suited for any possible domain.

## Planning Evolution

After each IPC, certain techniques have risen as the *state-of-the-art*. In the past, heuristic search was most of the time the best approach, and certain heuristics were highly successful (Helmert and Domshlak 2009). Symbolic search has also had success with SymBA*, a symbolic bidirectional planner (Torralba et al. 2014).

Each winner of the competition has shown the planning community which combination of technique and domain works especially well. Most the best performing planners have been awarded more attention in the following years, bringing forth their ideas in the community. Also, each well performing planner in the competition has made the organizer of the following competition to make their benchmark set harder for those techniques. This has made the community now to have a very diverse set of problems on which we can see how well each planner performs.

Portfolio planning is a technique that tries to combine find the best planner for the domain/problem that it has to solve. Work done by Sievers et al. (2019) has shown how grounded problems can be classified as to give the *better suited* planner the problem to solve. Following this work, portfolio planners are aiming to find the planners best suited for a type of planning task. This is a different approach to domain-independent planning, but has shown good results when looking at the results from the 2018 IPC.

## 3 Measuring Cost-Optimal Planning

In this section we will be discussing different ways of comparing planners, and see how they differ from each other. We have tested five planners, Complementary 1 and 2, Planning-PDBs, Scorpion and Symbolic-Bidirectional on a 69 domains, all the benchmarks from the previous three competitions and a subset of the domains from before 2011.

## Coverage

As stated in the first section, the current way of comparing cost-optimal classical planning is by measuring the coverage of a planner (i.e. how many problems a planner can solve on a set of problems). Each problem solved is counted as a point towards that planner and at the end we compare the tally of each planner, the one with the most being the winner.

This metric is used both in competitions and in published papers when measuring the performance of a new method. However, this metric can become domain dependant if the number of problem instances is not uniform over all the domains. In our pre-2011 set of problems, made out of 31 domains, we can see that some domains are a lot more important than others when using this approach (seen in figure 1).

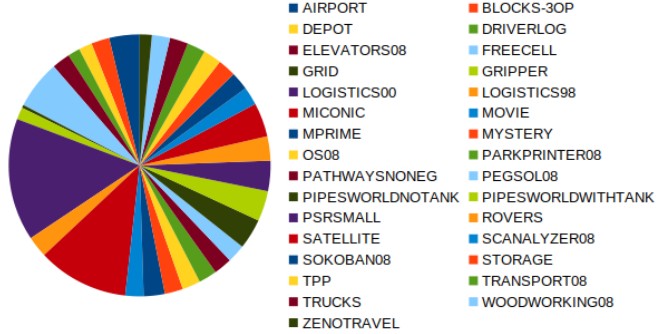

Figure 1: Size of domains from our pre-2011 benchmark

The benchmarks from 2011 and 2018, the domains were kept at a uniform size of 20 instances each. In that case, there is no need to normalize the results, but when using becnhmark sets like 2014 (most had 20 instances, with three different) and the pre-2011 we used (from 5 to 202, with most having 30 instances), the change in domain sizes requires a change in the evaluation metric.

| | Problems Solved | Coverage | Normalized Coverage |
|---|---|---|---|
| Planning-PDBs | 1122 | 54.17% | 59.42% |
| Complementary1 | 1099 | 53.06% | 57.60% |
| Complementary2 | 1164 | 56.15% | **62.08%** |
| Scorpion | **1208** | **58.32%** | 60.11% |
| Sym-BiDir | 1053 | 50.84% | 55.46% |

Table 1: Overall results as number of problems solved, coverage and normalized coverage.

## Normalized Domain Coverage

For cases like this, we normalize the domain coverage, and then get the average for each planner. By doing this, we

| | Pre 2011 | Coverage | Normalized Coverage | IPC11 | Coverage (also Normalized) | IPC14 | Coverage | Normalized Coverage | IPC18 | Coverage (also Normalized) |
|---|---|---|---|---|---|---|---|---|---|---|
| Planning-PDBs | 678 | 50.78% | 55.88% | 190 | 67.85% | 131 | 51.17% | 53.48% | 123 | 61.5% |
| Complementary1 | 680 | 50.93% | 55.95% | 185 | 66.07% | 111 | 43.35% | 46.15% | 123 | 61.5% |
| Complementary2 | 686 | 51.38% | 56.95% | **198** | **70.7%** | **155** | **60.54%** | **61.99%** | **124** | **62%** |
| Scorpion | **785** | **58.80%** | **60.20%** | 190 | 67.85% | 118 | 46.09% | 48.77% | 104 | 52% |
| Sym-BiDir | 647 | 48.45% | 53.46% | 174 | 62.14% | 129 | 50.39% | 52.97% | 114 | 57% |

Table 2: Results of the five planners on the pre-2011, IPC11, IPC14 and IPC 18 benchmarks. For each benchmark we have the number of problems solved, coverage and normalized coverage (where needed).

first see how much of a domain a planner can solve, and then by averaging we get a better metric for overall domain-independent performance of a planner.

We can see the value of such a metric in table 1, where we can see that, even though Scorpion solves the most probles out of the total of 2071 we tested on, the normalized coverage is worse than the one of Complementary2 (62.08% to 60.11%).

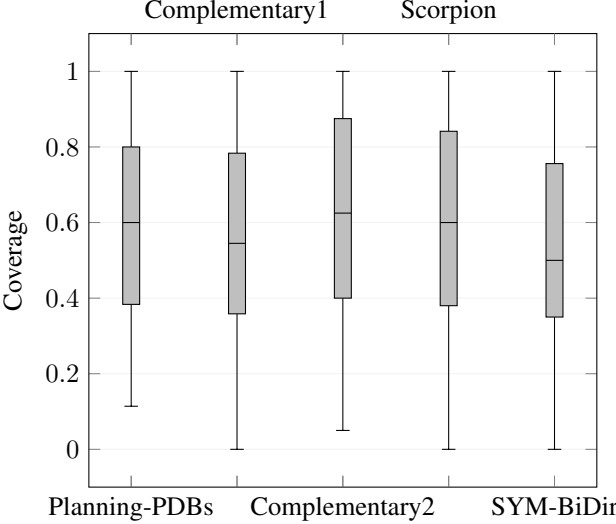

By looking at the boxplot of the normalized per-domain coverage of each planner, we can see an even clearer picture. The only two planners to solve problems on all the 69 domains are Planning-PDBs and Complementary2. Also, Planning-PDBs is a lot closer to Scorpion than what the number of instances solved would imply (1208 to 1122).

Another way of measuring the perfocmance when having a normalized coverage, would be by getting the median value for each planner. In our test cases, we find that Complementary2 has the best with 65%, with Scorpion and Planning-PDBs following (both have 60%). Complementary1 and Symbolic-Bidirectional finish the top with 55% and 50%.

## 4 Future suggestion

We can see that having the same number of problem instances for each domain is vital for evaluating domain independence. The organizers of IPC11 and IPC18 saw this and had an uniform number of problems. But in the future, by using a normalized domain coverage, future organizers

can break from this constraint. Some domains would need a more granularity to differentiate the planners (domains where most planners get the same coverage). While organizing the competition and seeing this, organizers can add more instances for those domains.

Also, each IPC has contributed with new domains and problems that have been added by the community for evaluating their planners and subsequently adding them to their results sections in conference and journal publications. From just a glance at the problems we evaluated on, we can identify that from the current number of available domains there are more instances per domain from before IPC11, with an average of 30, and since 2011 having an average of 20. This will make in any evaluation section the domains from before 2011 of a more importance and not have a fair comparison of the domain-independence of a planner.

## 5 Conclusion

In this short paper, we propose that we should compare planners not only by the number of problems solve, but also by normalized pre-domain coverage as to evaluate the domain independence of a planner.

We do not want to subtract any value from the previous method. Solving more problems will always be a great indicator to the performance of the planner. However, due to the nature of our current set of problems, we identify that the variance in number of instances per domain is an issue and could lead to planner developers focusing their attention to solving the domains with the most instances.

We do not touch on any other metric of evaluating a certain technique in cost-optimal planning. There are many other ways that we can use to show an even more complete picture of a planner, but that is for future work.

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
