# OpenReview forum: "Benchmarks Old and New: How to compare domain independence for cost-optimal classical planning?"
_icaps-conference.org/ICAPS/2019/Workshop/WIPC_

### Official Review · AnonReviewer3 · 2019-04-17
**Weighting coverage by domain size is important but not a new idea**

**Rating:** 6
**Confidence:** 5

**Review:**

The paper reminds us to report coverage results weighted by domain
size. It also contains a short history of planning and the IPC which is
unrelated to the paper's main message that plain coverage results can
be skewed towards large domains.

While I completely agree that coverage results should be weighted by
domain size, this is nothing new. It has been used like this papers for
a while now (usually only when the results show a particular bias
towards or against a large domain). It is also the reason why the IPC
uses equally sized domains since 2011. (Yes, the 2014 domains are not
equally sized but they were designed as equally sized for the
competition and instances had to be removed afterwards.) The score has
been used by different names in papers, such as "coverage score" (to
distinguish it from "total coverage"), "normalized coverage", or "the
average probability to solve a task from a randomly selected domain".
For example, Richter and Hemert (ICAPS, 2009) do this by first
computing a score for each domain and then reporting the average of the
domain scores. It is somewhat hard to search for this because (in my
mind) this is not a new metric but just basic rules of statistics: if
data that comes from an uneven distribution is aggregated, it should be
weighted according to the distribution.

Anyway, it could be good to repeat this in the workshop but I doubt
that it needs a 15 minutes slot. For the paper, I would prefer if it
focused more on the main message (I don't see what Section 2
contributes to this message and suggest to remove it). It also should
acknowledge that this metric is not new and is already in use. I'm not
claiming that Richter and Helmert invented the metric, I'm saying that
it is just the normal way of dealing with such data.

As for the point that future IPCs could make use of this to have
differently sized domains: I disagree that this is something that IPC
organizers should go for. If a selection of 20 instances is not
sufficient to differentiate planners but a selection of 30 instances
is, then a different subset of 20 tasks from the 30 should be
sufficient to differentiate the planners as well and the remaining 10
tasks are then too similar to other tasks in the set. Finding a set of
20 instances that is well-spaced in the difficulty space between
trivial-to-solve and impossible-to-solve is hard but it is the job of
the organizers. Adding the option of using more tasks will just lead to
benchmark sets that contain too many tasks of similar difficulty.

Finally, the problem of differently-sized domains is only one of the
problems when papers aggregate coverage over multiple domains: the
benchmark set now contains several domains that are copies or slight
variations of previously used domains. Weighting by domain size helps
to remove the bias for domains like miconic (wih 150 instances) but it
also increases the bias for domains like openstacks (that was used in
IPCs 2006, 2008, 2011, and 2014, sometimes with identical instances).
Adding a discussion on this topic would improve the paper as well.


Some typos:
* missing article before current in the abstract
* a better picture *of*
* no comma before and at the start of section 3
* perfocmance -> performance (please use a spell checker)
* conclusion: solve*d* but also by normalized *per*-domain coverage
* indicator *of*

---

### Official Review · AnonReviewer2 · 2019-04-24
**Normalizing coverage results is reasonable if domains with different number of instances are used**

**Rating:** 6
**Confidence:** 4

**Review:**

The main message of this paper is based on the observation that coverage
results can be biased if domains with a varying number of instances are
used.

While this is a valid observation, I would be quite surprised if it has
not been observed before. I am not an expert in classical planning,
though, and couldn't find an example in the area where it has been used.
However, at IPC 2011 and 2014, the probabilistic tracks of IPC were
evaluated with the same metric that is proposed here (even though it was
not necessary there as all domains had the same number of instances).
Anyway, as long as there hasn't been a specific publication on this
topic, I don't see a reason not to state this insight explicitly.

The main weakness of the paper is the fact that it is not very focused
and kind of blown up to deliver a simple message:

- I am not sure what the subsection on "Planning Evolution" has to do
  with the rest of the paper. Please remove that part or make sure it is
  clear why it is relevant.

- I don't see why Table 2 is relevant to argue for the normalized
  coverage. In my opinion, Table 1 is sufficient to backup your claim.

- Similarly, why is the boxplot figure relevant? Do you argue that we
  should use median per-domain results instead of (or in addition to)
  normalized coverage? And why is it relevant that some planners solve
  problems from all domains? Do you propose that that should also
  influence the result of future IPCs? It is not clear to me why you
  include this.

I would really like this paper if it just consisted of the introduction,
the first part of Section 3 up to the boxplot figure, Section 4 and the
conclusion. For the other parts, I'd like to see why they are relevant
to the topic.